# Learning from Rich Semantics and Coarse Locations for Long-tailed Object Detection

**Lingchen Meng**[1,2]    **Xiyang Dai**[3]    **Jianwei Yang**[3]    **Dongdong Chen**[3]    **Yinpeng Chen**[3]
**Mengchen Liu**[3]    **Yi-Ling Chen**[3]    **Zuxuan Wu**[1,2†]    **Lu Yuan**[3]    **Yu-Gang Jiang**[1,2]

[1]Shanghai Key Lab of Intell. Info. Processing, School of CS, Fudan University
[2]Shanghai Collaborative Innovation Center of Intelligent Visual Computing
[3]Microsoft

## Abstract

Long-tailed object detection (LTOD) aims to handle the extreme data imbalance in real-world datasets, where many tail classes have scarce instances. One popular strategy is to explore extra data with image-level labels, yet it produces limited results due to (1) **semantic ambiguity**—an image-level label only captures a salient part of the image, ignoring the remaining rich semantics within the image; and (2) **location sensitivity**—the label highly depends on the locations and crops of the original image, which may change after data transformations like random cropping. To remedy this, we propose RichSem, a simple but effective method, which is robust to learn rich semantics from coarse locations without the need of accurate bounding boxes. RichSem leverages rich semantics from images, which are then served as additional "soft supervision" for training detectors. Specifically, we add a semantic branch to our detector to learn these soft semantics and enhance feature representations for long-tailed object detection. The semantic branch is only used for training and is removed during inference. RichSem achieves consistent improvements on both overall and rare-category of LVIS under different backbones and detectors. Our method achieves state-of-the-art performance without requiring complex training and testing procedures. Moreover, we show the effectiveness of our method on other long-tailed datasets with additional experiments. Code is available at https://github.com/MengLcool/RichSem.

## 1   Introduction

Object detection for complex scenes has advanced significantly [10, 9, 40, 3, 65] thanks to large-scale datasets [5, 22, 32, 12, 24, 4, 42]. However, current deep models depend on relatively balanced large-scale datasets, where different classes have similar numbers of images and samples, to learn diverse semantics and enough locations. This limits their performance on real-world data, which are often long-tailed, *i.e.*, only a few head classes have plenty of training samples while most classes have very few training samples, making detection more challenging and less effective for rare classes.

A simple and effective way to improve long-tailed object detection (LTOD) is to use extra data to increase the training samples for tail classes. However, collecting bounding box annotations, especially for rare categories, is costly and tedious. Therefore, previous studies resort to datasets with image-level labels to enrich the amount of samples for rare classes by exploring image-level semantics (as shown in Figure 1 (a)). While appealing, directly learning from such data to benefit detection is challenging since they lack bounding box annotations that are essential for object detection. To remedy this, many works [66, 25, 60, 63] focus on estimating bounding boxes for

---

[†] Corresponding author.

37th Conference on Neural Information Processing Systems (NeurIPS 2023).

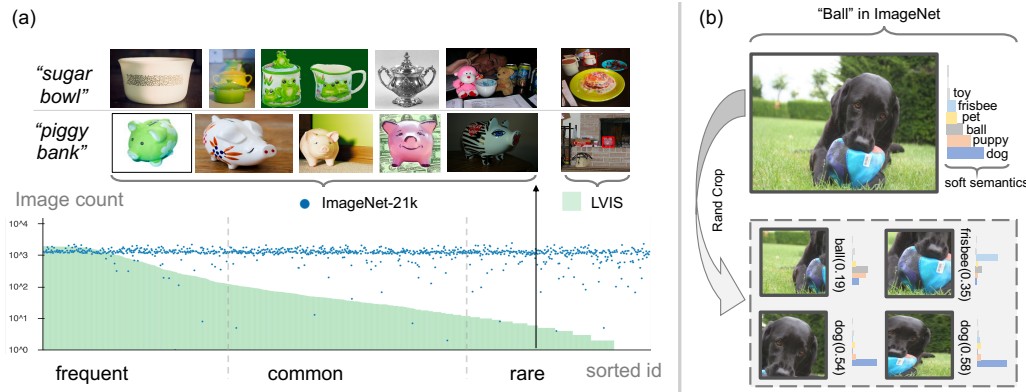

Figure 1: **(a) Category occurrences in the long-tail detection data (LVIS) and the extra classification data (ImageNet-21k).** The green bars show the number of images that each category occurs in LVIS [12]; the blue dots show the number in ImagetNet-21k [5]. Evidently, the classification dataset exhibits a more evenly balanced distribution of occurrences. Moreover, the classification dataset offers a broader range of instances, thereby enhancing the diversity of instances for each category. **(b) A sample from ImageNet [5].** This sample is annotated as "ball" while ignoring another main object "dog". After a random crop during training, the "ball" may be inaccurate on some crops. The class with the highest CLIP confidence score is shown above each crop.

objects in images. They typically consider image-level tags (in the form of one-hot labels) as the ground-truth classes and match with best estimated boxes as pseudo labels for training.

We argue that image-level labels are not well-suited for detection due to their *semantics ambiguity* and *location sensitivity*. On one hand, an image-level label can not sufficiently reflect all the semantics within the image. On the other hand, the label mainly focuses on the iconic object; thus, using a different crop might shift the semantics to a different object. Take Figure 1 (b) as an example: the top image is labeled as "ball" in ImageNet [5], which only depicts part of information in the image. The panel below in Figure 1 (b) shows that the provided image-level label is no longer accurate after data augmentations. Inspired by the recent success of visual-language contrastive models that align a large number of image-text pairs [37, 62, 11, 54, 13], we aim to leverage such models to extract rich semantics that are more informative than the image-level labels in classification datasets. However, naively converting one-shot labels in image classification tasks to a distribution of soft labels is still not optimal and accurate after data augmentation. As shown in Figure 1 (b), based on different locations (*i.e.*, random crops) of the image, the semantics extracted by CLIP [37] incur more noise than the original image.

To address this, we introduce RichSem, a one-stage training framework that leverages additional image data to boost the detector through learning from *rich semantics* and *coarse locations* for long-tailed object detection. In particular, we treat a whole-image box as coarse locations (*i.e.*, the coarse bounding box shares the same size as the image) and group multiple images together to build a mosaic. We then use CLIP to extract the semantics according to such coarse bounding boxes. The extracted semantics can serve as "soft labels" to enrich the amount of training data and are more robust to random cropping. We further introduce a new branch named as "semantic branch" to object detectors so that it can learn from the derived semantics during training. This allows the detector to fully leverage the rich semantics in classification datasets with minimal modifications. Once trained, this branch is discarded during inference, and RichSem can be used readily as a standard detector.

Our contributions are summarized as follows: (1) We point out that using image-level labels from extra classification datasets as supervision is challenging due to *semantics ambiguity* and *location sensitivity*, which can be alleviated with a distribution of semantics serving as "soft labels". We show that semantics provided by CLIP are not only rich and but also robust to locations, providing better guidance than original image-level labels. (2) We introduce a novel semantics learning framework named RichSem, which uses an additional branch to learn from *rich semantics* and *coarse locations* for long-tailed object detection without the need to compute pseudo labels. Once trained, the additional branch can be discarded during inference. (3) Our method demonstrates strong results on long-tailed datasets, *e.g.* LVIS, highlighting that it is a low-cost way to boost the detector performance for long-tailed object detection by only complementing extra classification data.

## 2 Related Work

**Long-tailed object detection (LTOD)** has attracted more and more attention. The performance of rare categories is drastically low compared to frequent categories due to the unbalanced distribution and the lack of training samples. Existing works can broadly be divided into two directions: 1) one direction aims to improve the training scheme for balanced learning, including data re-sampling [12], loss re-weighting [46, 45, 51], data augmentations [8] and decoupled training [23, 21]; 2) Another direction leverages extra data to compensate for the data starvation [63, 60]. These methods often trust image-level labels to boost the classification capability. We find that it is far from optimal to naively treat image-level labels as golden labels and supervision for the classifier. Unlike existing methods, we study a new solution that leverages the *rich semantics* within the images from classification data with only *coarse locations* provided by regular augmentations. We leverage CLIP to provide semantics as soft targets on classification data to guide our detector to learn semantics explicitly. In this embarrassingly simple but effective way, we can better leverage classification data for LTOD.

**Weakly-supervised object detection** aims to train a detector using image-level labels without bounding boxes. Many studies [1, 47, 36] train a model using only image-level labels without any bounding box supervision. Another line of work [39, 60] takes the bounding boxes supervision with the whole-image labels together under a semi-supervised framework. Unlike prior works, we focus on mining rich semantics within the images instead of bounding box estimation. Thus, we no longer need to estimate precise bounding boxes on classification data.

**Language supervision for object detection** is a recent topic that aims to leverage linguistic semantics. Since language supervision has rich semantics, each category is related rather than independent in one-hot labels. Thanks to this property, recent works [6, 59, 57, 20, 35, 55] show benefits by pre-training backbones on vision-language tasks. With the rapid progress in contrastive language-image pre-training [37, 18], many recent approaches [28, 11, 62] apply large-scale pre-training for object detection. ViLD [11] and RegionCLIP [62] attempt to align the visual-semantic space of a pre-trained CLIP model for open-vocabulary detection. Similar to RegionCLIP and ViLD, our method leverages the visual-semantic space learned from pre-trained CLIP models. In contrast, our goal is to leverage object semantics to boost object classification, especially for rare categories. Since CLIPs are used to generate "soft labels", our backbone is free of CLIP initialization compared with RegionCLIP [62].

**Knowledge distillation and soft label.** Knowledge distillation (KD) [17] is a powerful tool to boost the student model with prior knowledge of the teacher model. Recent follow-ups extended and developed many variants, *e.g.*, feature distillation [19, 41], hard distillation [50], contrastive distillation [48], etc. To improve the training efficiency of KD, Re-label [58] and FKD [44] use a strong teacher model to generate soft labels and store them for efficient image recognition training. Recently, many studies [11, 34, 62, 38] introduce knowledge from CLIP [37] into the KD framework to boost open-vocabulary object detection. Due to the strong semantics capturing capability of CLIP, those methods show encouraging performance on open-vocabulary detection. Similar to those works, our approach leverages the semantic knowledge of pre-trained CLIP models. In contrast, our goal is to boost the long-tailed detection, especially for the tailed classes, with the help of rich semantics of classification and detection datasets. Moreover, we introduce a simpler but more effective way to learn visual semantics with an extra semantic branch during training. Unlike [62, 11], our one-stage training scheme needs no fine-tuning after pre-training. Besides, our approach is more effective since there are no redundant pre-computed boxes [11] and boxes estimation [34, 38].

## 3 Method

Given a detection dataset denoted as $\mathcal{D}^{od} = \{(I^{od}, \{(b^{od}, c^{od})\})\}$, where each image $I^{od}$ is associated with bounding boxes $b^{od}$ and class labels $c^{od}$ respectively. Our goal is to leverage an additional classification dataset $\mathcal{D}^{extra} = \{(I^{img}, c^{img})\}$, of which each image is only labeled with an image-level label $c^{img}$, to improve long-tailed object detection, particularly for rare classes.

In long-tailed object detection, previous approaches typically rely on image-level labels and compute pseudo boxes for corresponding objects. On one hand, the semantic information provided by such labels is limited, (*i.e.*, one one-hot label per image). On the other hand, augmentation strategies like randomized cropping might generate regions that do not contain the annotated class, thus making the provided labels no longer accurate. Motivated by the fact that CLIP has strong capabilities of

capturing visual semantics conditioned on only coarse locations [1] [62], we build upon CLIP to explore image classification datasets to guide the training of object detectors. Below, we first introduce how to obtain rich semantics from CLIP in Section 3.1, and the resulting semantics are then used to guide the training of our detector, as will be described in Section 3.2. Then we describe how we unify the training objective and extend RichSem to different types of extra datasets in Section 3.3.

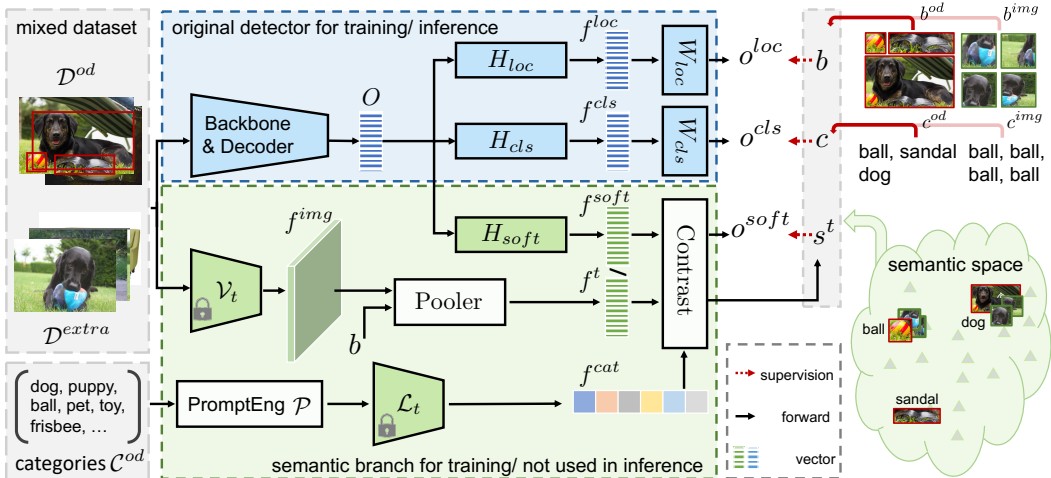

Figure 2: **Our RichSem framework.** $\mathcal{V}_t$ and $\mathcal{L}_t$ indicate vision encoder and text encoder of the CLIP model. We leverage rich semantics in extra data with image-level labels by pre-trained CLIP. Then semantics behave like "soft labels" for our proposed semantic branch ($H_{soft}$ followed by Contrast). Blue parts indicate the components for traditional object detection and green parts indicate those for semantics learning.

## 3.1 Image and Object Semantics

We aim to use image-level labels in classification datasets together with detection datasets to improve long-tailed object detection. This requires processing these two types of data within the same framework. As a result, we treat images from detection and classification datasets equally and we not only derive semantic information at the image-level for classification datasets but also at the object-level for detection datasets, as will be elaborated below.

**Image-level semantics.** We denote the pre-trained CLIP visual encoder and the language encoder as $\mathcal{V}_t$ and $\mathcal{L}_t$, respectively. Consider an image $I$ after random cropping, its semantics $s$ can be obtained by computing the similarity between its visual features $f^{img}$ extracted by the visual extractor and the linguistic categories features $f^{cats}$ produced by the language encoder. Formally, we compute the visual and linguistic features as follows:

$$f^{img} = \mathcal{V}_t(I^{img}); \quad f^{cats} = \mathcal{L}_t(\mathcal{P}(\mathcal{C}^{od}))$$
$$s^{img} = \text{Contrast}(f^{img}, f^{cats}). \tag{1}$$

Here, $\mathcal{P}$ is the prompt engineering function to convert class names into prompts; Contrast is the contrasting paradigm [37] to calculate the similarity between two features, and $s^{img}$ contains semantics information in which each element indicates the likelihood for the corresponding class. Following [37, 62], we use a set of prompts to convert the categories in the target vocabulary $\mathcal{C}^{od}$ into sentences then use the CLIP text encoder $\mathcal{L}_t$ to obtain the linguistic features of categories in $\mathcal{C}^{od}$.

**Object-level semantics.** As mentioned above, to unify the training process with classification and detection data, we also obtain object-level semantics from CLIP as well. To this end, we obtain an object-level representation by pooling from image-level features [9, 15] rather than cropping in the original images, following [62]. More formally, object-level semantics are obtained as:

$$f^t = \text{Pooler}(\mathcal{V}_t(I), b)$$
$$s^t = \text{Contrast}(f^t, f^{cats}) \tag{2}$$

---

[1]Please refer to our supplementary material for the verification of coarse location

where Pooler is RoIAlign [15] that pools region features according to their locations from the entire feature maps, and $b$ is the location of the object, which could be ground-truth, predicted, or predefined whole-image bounding boxes.

It is worth pointing out with such a formula both detection and classification datasets can now be unified, as objects in detection datasets are essentially regions of images, while images from classification datasets are cropped regions from original images. Consequently, the $b$ in Equation (2) indicate tight ground-truth bounding boxes $b^{od}$ for objects in detection datasets, while $b$ corresponds to the entire image, *i.e.*, coarse whole-image boxes $b^{img} = (0, 0, h, w)$, for samples from classification datasets, where $h, w$ is the height and width of the augmented image.

Furthermore, unlike previous approaches that use fixed and static semantics provided by image-level labels, we obtain image-level and object-level semantics which dynamically change conditioned on different locations ( *i.e.*, bounding boxes) in an online fashion. This is particularly useful when location information is not accurate—the location could be a very loose bounding box for the object of interest or even contains part of the objects. As as result, the rich and location-robust semantics can be used to guide the detector.

## 3.2 Semantics as Soft Labels

Now that we obtain both image-level and object-level semantics with CLIP, the main challenge is how to make our detector learn from them effectively. Traditional detection models [40, 31, 2, 3, 65] usually obtain object features from whole-image features according to locations [9, 40, 31, 49] or cross-attention [3, 65]. Then object features are refined with a classification branch to classify the region into predefined categories, and a location branch to compute the bounding box, separately. To unify the notations, we denote the object features as $O$, which are fed to different branches to obtain features tailored for each sub-task to generate final outputs with a projection layer. More formally,

$$
\begin{aligned}
f^{loc}, f^{cls} &= H_{loc}(O), H_{cls}(O) \\
o^{loc}, o^{cls} &= W_{loc}(f^{loc}), W_{cls}(f^{cls}),
\end{aligned}
\tag{3}
$$

where $H$ denotes the sub-task branch for feature refinement, specifically $H_{loc}$ is for localization and $H_{cls}$ is for classification; $W_t$ denotes the projection layer to produce final results; $f_t$, and $o_t$ denotes the features and outputs for the corresponding task, *i.e.*, $t \in \{loc, cls\}$.

Now we discuss how to use the obtained semantics to guide the training of detectors. As $o^{cls}$ are logits that are generally used for classification, a straightforward way is, on top of a cross-entropy loss, to use $o^{cls}$ to predict the object semantics $s^t$ in a similar spirit to knowledge distillation [17, 41]. However, this makes the training of detectors challenging as the cross-entropy loss and distillation loss have conflicting purposes: a cross-entropy loss is generally optimized to produce hard prediction results while the distillation loss aims to borrow knowledge from a distribution of semantic scores that serve as "soft labels". As a result, jointly performing hard predictions and soft distillations conditioned on $o^{cls}$ produce unsatisfactory results, as will be shown empirically.

To mitigate the challenge, we introduce an additional branch named as "semantic branch", independent of the classification and localization branches in current detectors, to learn the soft semantics obtained by CLIP. The semantic branch also includes a feature refinement branch $H_{soft}$ then feed the refined feature $f^{soft}$ to Contrast function to obtain the semantic prediction $o^{soft}$. Such a strategy ensures that semantics from image classification datasets can be explored without interfering with the original training process of detectors. This is essentially using semantics as soft teachers to implicitly refine object features for detection. Formally, to train the semantic branch, we use a KL divergence loss as:

$$
\begin{aligned}
o^{soft} &= \text{Contrast}(f^{soft}, f^{cat}), \quad f^{soft} = H_{soft}(O) \\
L_{soft} &= \frac{1}{N} \sum_{i=1}^{N} L_{KL}(o_i^{soft}, s_i^t)
\end{aligned}
\tag{4}
$$

where $o_i^{soft}$ is the semantics prediction of the $i$-th object feature, while $s_i^t$ is the corresponding semantic target; $N$ is the number of matched proposals/ queries during training. It is worth noting that we only use the semantic branch for distilling semantics into the detector during training. And once trained, the semantic branch is no longer needed for inference.

### 3.3 Unified Objective Functions

We aim to incorporate semantic learning from classification datasets into an end-to-end learning scheme instead of fine-tuning after pre-training [62], without redundant pre-computed boxes [11] and box estimation [34, 38]. Thus, we unify the objective functions and treat datasets of classification and detection equally in a unified way.

In our training scheme, we use a unified classification loss $L_{unicls}$, the combination of a hard classification loss and a soft semantics loss, to boost the capability of object classification. The hard classification loss $L_{cls}$ best works when bounding boxes and class labels are available and sufficient for training. In contrast, the soft semantic learning loss $L_{soft}$ works well for those categories with few samples, requiring only coarse locations.

$$L = \lambda_{loc} \cdot L_{loc} + \underbrace{\lambda_{cls} \cdot L_{cls} + \lambda_{soft} \cdot L_{soft}}_{L_{unicls}} \tag{5}$$

where $\lambda_{loc}$, $\lambda_{cls}$ and $\lambda_{soft}$ means the weight of each loss.

Furthermore, the vocabularies of target detection datasets and extra classification datasets often differ. To handle the taxonomy difference between detection and classification datasets, we further unify the supervision for hard classification as follows.

**Handling taxonomy differences.** To leverage image-level labels in extra data, we need to map the vocabulary of extra data $\mathcal{C}^{extra}$ to the vocabulary of our target detection dataset $\mathcal{C}^{od}$, which is a function $\mathcal{M} : \mathcal{C}^{extra} \rightarrow \mathcal{C}^{od}$. One could derive better mapping functions $\mathcal{M}$, yet this is orthogonal to our current direction. For convenience, we can use a manual label mapper or leverage the CLIP text encoder to map the class according to their category semantics automatically.

$$c^{od} = \mathcal{M}(c^{img}) \tag{6}$$

**Handling data without any labels.** We also consider a broader case, in which even image-level labels are not available. Inspired by DeiT [50], we treat the class of highest logits in each semantics target as the label for classification. We also filter $c^{hard}$ with a threshold to filter out those images relevant to the vocabulary of the target set.

$$c^{hard} = \begin{cases} \arg\max(s^t) & \text{if conf} > th \\ \varnothing & \text{else} \end{cases} \tag{7}$$

where $c^{hard}$ is the generated class label for hard classification.

## 4 Experiments

We conduct experiments and analysis on the task of long-tailed object detection. We mainly evaluate our method on LVIS [12] val 1.0 for our main experiments and ablations. We also conduct experiments on other datasets of long-tail distribution to further prove the effectiveness. We use DINO [61], a advanced DETR-based detector due to the training efficiency and high performance.

### 4.1 Datasets and evaluation metrics.

**Long-tail detection dataset.** We mainly conduct experiments on LVIS [12], which contains 1203 classes with ∼100K images. The classes are divided into rare, common, and frequent groups based on the number of training images. The category distribution is extremely long-tailed since instances in rare categories are usually less than 10. Moreover, we experiment on other datasets, *e.g.* Visual Genome [22] and OpenImages [24], please refer to our supplementary material.

**Extra data.** We mainly use ImageNet-21k [5] as additional classification data. ImageNet-21k consists of ∼14M images for 21K classes, and there are 997 classes that overlap with the vocabulary in LVIS. We follow the label mapping used in Detic [63] and denote the subset of overlapped classes as ImageNet-LVIS, which contains ∼1.5M images with 997 LVIS classes. For the full set, we treat the data as unlabeled images and leverage our learning scheme to learn from it. We also treat the

| Method | Backbone | Schedule | AP | $AP_{50}$ | $AP_{75}$ | $AP_r$ | $AP_c$ | $AP_f$ |
|---|---|---|---|---|---|---|---|---|
| DINO*[61] | R50 [16] | 1× | 28.8 | 38.4 | 30.3 | 18.2 | 26.5 | 36.1 |
| MosaicOS† [60] | R50 [16] | 1× | 25.0 | 40.8 | 26.5 | 20.2 | 23.9 | 28.3 |
| Detic-DDETR [65] | R50 [16] | 4× | 31.7 | - | - | 21.4 | 30.7 | 37.5 |
| Detic-DDETR† [63, 65] | R50 [16] | 4× | 32.5 | - | - | 26.2 | 31.3 | 36.6 |
| Detic-CenterNet2 [63] | R50⋆ [16] | 4× | 35.3 | 48.7 | 37.2 | 28.2 | 33.8 | 40.0 |
| Detic-CenterNet2† [63] | R50⋆ [16] | 4× | 36.8 | 50.7 | 38.6 | 31.4 | 36.0 | 40.1 |
| RichSem (Ours) | R50 [16] | 1× | 32.2 | 42.3 | 33.9 | 24.1 | 29.9 | 38.3 |
| RichSem (Ours) | R50 [16] | 2× | 34.9 | 45.5 | 36.6 | 26.4 | 32.5 | 41.3 |
| RichSem (Ours) | R50 [16] | 3× | 35.1 | 45.8 | 36.8 | 26.0 | 32.6 | 41.8 |
| RichSem ‡ (Ours) | R50 [16] | 1× | 35.0↑2.8 | 46.0 | 36.7 | 30.4↑6.3 | 33.1 | 39.0 |
| RichSem ‡ (Ours) | R50 [16] | 2× | 37.1↑2.2 | 48.2 | 39.0 | 29.9↑3.5 | 35.6 | 42.0 |
| RichSem ‡ (Ours) | R50⋆ [16] | 2× | 40.1↑5.2 | 51.9 | 42.3 | 36.2↑11.8 | 38.2 | 44.0 |
| RichSem (Ours) | Swin-T [33] | 1× | 34.9 | 45.5 | 36.9 | 26.0 | 32.6 | 41.3 |
| RichSem ‡ (Ours) | Swin-T [33] | 1× | 38.3↑3.4 | 49.8 | 40.4 | 34.1↑8.1 | 36.3 | 42.3 |
| RichSem (Ours) | Swin-T [33] | 2× | 38.8 | 49.9 | 41.0 | 30.8 | 36.4 | 45.0 |
| RichSem ‡ (Ours) | Swin-T [33] | 2× | 41.6↑2.8 | 53.3 | 43.8 | 37.3↑6.5 | 39.7 | 45.5 |
| Detic-CenterNet2 [63] | Swin-B [33] | 4× | 45.4 | 59.9 | 47.9 | 39.9 | 44.5 | 48.9 |
| Detic-CenterNet2†[63] | Swin-B [33] | 4× | 46.9 | 62.2 | 49.4 | 45.8 | 45.5 | 49.0 |
| RichSem (Ours) | Swin-B [33] | 2× | 46.4 | 59.2 | 48.9 | 38.5 | 45.1 | 51.3 |
| RichSem ‡ (Ours) | Swin-B [33] | 2× | 48.2↑1.8 | 61.6 | 51.0 | 46.5↑8.0 | 46.5 | 51.0 |
| ViTDet [29] | ViT-L-MAE [14] | ∼8× | 51.2/ 49.2 | - | - | - | - | - |
| ViTDet [29] | ViT-H-MAE [14] | ∼8× | 53.4/ 51.5 | - | - | - | - | - |
| RichSem (Ours) | Swin-L [33] | 1× | 47.0 | 59.9 | 49.6 | 41.2 | 45.9 | 50.7 |
| RichSem ‡ (Ours) | Swin-L [33] | 1× | 49.8↑2.8 | 63.7 | 52.5 | 48.6↑7.4 | 49.7 | 50.5 |
| RichSem (Ours) | Swin-L [33] | 2× | 49.7 | 62.9 | 52.4 | 42.8 | 49.2 | 53.4 |
| RichSem ‡ (Ours) | Swin-L [33] | 2× | 52.0↑2.3 | 65.7 | 54.8 | 50.2↑7.4 | 51.5 | 53.3 |
| RichSem (Ours) | Focal-L [56] | 1× | 49.5 | 62.5 | 52.2 | 47.3 | 47.7 | 52.4 |
| RichSem ‡ (Ours) | Focal-L [56] | 1× | 51.5↑2.0 | 65.2 | 54.4 | 52.3↑5.0 | 50.7 | 52.0 |
| RichSem (Ours) | Focal-L [56] | 3× | 51.4 | 64.8 | 54.2 | 47.6 | 49.8 | 54.9 |
| RichSem ‡ (Ours) | Focal-L [56] | 3× | 53.6↑2.2 | 67.2 | 56.7 | 52.8↑5.2 | 52.6 | 55.2 |

Table 1: **Results on LVIS val v1.0.** * indicates that we train the model on LVIS using their official hyper-parameters best trained on COCO; R50 indicates ResNet50 [16] pre-trained on ImageNet-1k, while R50⋆ indicates pre-trained on ImageNet-21k; RichSem (Ours) indicates that we train our baseline model under the original object detection dataset with traditional loss; RichSem ‡ (Ours) indicates using additional extra data under our training scheme. Besides, †indicates that the model is trained with extra data. In this table, our models use the subset of ImageNet-21k, of which categories overlap with LVIS vocabulary as extra dataset. The AP of ViTDet includes the AP reported paper (the former) and the AP reported in their official repo (the later).

full set as unlabeled dataset ignoring the image-level labels, denoted as INet-Unl. Furthermore, we explore an additional detection dataset Object365 [42] and an image-text pair dataset CC3M [43].

**Evaluation metric.** We report box AP on LVIS with the LVIS official evaluator, which also contains $AP_r$, $AP_c$, $AP_f$ for rare, common and frequent class, respectively. We mainly focus on the performance gain on overall AP and AP of rare categories.

## 4.2 Implementation details

**Baseline model.** We take DINO [61], a powerful DETR-based detector with advanced query denoising [27] as our baseline. Differently, we convert the object classification into vision-language contrasting using the text features of categories.

**Mixed dataset training.** For experiments with extra data, we combine both detection datasets and extra datasets together to train our detector. We sample images from detection and classification datasets in a 1:1 ratio regardless of the original size of each dataset. Images in the same batch are sampled from the same dataset for high training efficiency. For mixed dataset training, we denote the schedule as the number of iterations on the target dataset for fair comparisons.

**Training details.** We adopt PyTorch for implementation and use 8×V100 GPUs. We set the initial learning rate as $1e$-4 and multiply 0.1 at the 11-th, 20-th and 30-th epoch for 1×, 2× and 3×, respectively, and set $\lambda_{soft} = 0.5$ for the soft semantics learning loss. Following [63], we use a federated loss [64] and repeat factor sampling [12] for LVIS; we use category aware sampling for OpenImages. We randomly resize an input image with its shorter side between 480 and 800, limiting

the longer size below 1333. Unlike training large models with larger scale images [29, 61], we use the same recipe for all models without any tricks or test time augmentation (TTA). For training with extra data, we use CLIP-RN50 to extract the semantic guidance for most models, while we use CLIP-RN50×16 [37] for Swin-L [33] and Focal-L [56] for better performance.

### 4.3 Main Results

**Results on LVIS.** Tab. 1 shows the result on LVIS. As shown in the table, detectors trained with our framework outperforms those trained with regular training recipes, especially for rare categories. When using a ResNet50 [16] as backbone networks, our baseline model is fully converged under $3\times$ schedule. Our RichSem under $2\times$ still outperforms the fully converged baseline, which demonstrates the effectiveness of our proposed method. Moreover, the experiments with Swin [33] backbones show consistent gains. Our model with a Swin-L as backbone achieves 52.0 AP and 50.2 AP$_r$, which outperforms the previous SoTA [29] by a large margin only under 1/4 training schedule. Overall, our RichSem makes the detector better on tailed categories and achieves significant gains on the corresponding AP, *e.g.* rare and common categories. Moreover, with better backbones, our RichSem achieves even balanced performance on rare, common and frequent categories. Notably, with a much smaller model size and

| Method | AP | AP$_r$ | AP$_c$ | AP$_f$ |
|---|---|---|---|---|
| Faster R-CNN [40] | 24.1 | 14.7 | 22.2 | 30.5 |
| EQL-v2 [45] | 25.5 | 17.5 | 23.9 | 31.2 |
| BAGS [30] | 26.0 | 17.2 | 24.9 | 31.1 |
| Seesaw Loss [51] | 26.4 | 17.5 | 25.3 | 31.5 |
| EFL [26] | 27.5 | 20.2 | 26.1 | 32.4 |
| MosaicOS ‡ [60] | 23.9 | 15.5 | 22.4 | 29.3 |
| CLIS ‡ | 29.2 | 24.4 | 28.6 | 31.9 |
| RichSem ‡ (Ours) | 30.6 | 27.6 | 29.7 | 32.9 |

Table 2: **Results on LVIS val v1.0 with Faster R-CNN as detector.** All models use R50 as backbone. '‡' indicates using extra classification data.

standard data augmentations for training and testing, our method rivals the ViTDet [29] that uses huge-sized model and sophisticated training recipes. We also compare compare the models with Faster R-CNN [40] as detector shown in Table 2. We further compare with large-scale vision foundation models [7, 53], please refer to our supplementary material.

### 4.4 Ablation Study

**Comparisons with CLIP initialization.** We replace the ResNet50 backbone with pre-trained CLIP visual encoder to study whether it can lead to balanced learning. As shown in Table 3a, although pre-trained CLIP has a strong ability to capture visual semantics, the detection performance even drops after finetuning on LVIS. The low AP$_r$ of CLIP backbone indicates that it is still biased towards the long-tailed distribution of downstream datasets. Therefore, using a **frozen** pre-trained CLIP to extract visual semantics as extra "soft supervision" can keep the generalization capability than building upon CLIP as initialization.

**Effectiveness of the semantics branch.** We study the importance of the proposed semantics branch by sharing the parameters of two heads instead of using two independent heads. As shown in Table 3b, when two heads share the parameters, the performance drops drastically and is even worse than the baseline model. This suggests that the objective of semantics learning differs from object classification. Therefore, semantics should be learned independently with the semantics branch.

**Effectiveness of rich semantics of classification data.** We conduct experiments using different types of extra data. Initially, we utilized Object365 [42], a detection dataset, and utilized its box annotations irrespective of the class labels. However, the performance gain was limited, as shown in Table 3c. This further verifies the assumption that the localization capability is not the primary bottleneck for long-tailed object detection [11]. Next, we experiment with ImageNet [5] and CC3M [43], two datasets with whole-image labels. We treat these datasets as unlabeled data and generate pseudo labels using Equation (7), denoted as INet-Unl and CC3M-Unl, respectively. As shown in Table 3c, utilizing these two datasets as extra data significantly outperformed the baseline, particularly for the rare categories. Furthermore, we employed ImageNet-LVIS, taxonomy mapped using Equation (6), as extra data, which yields the best performance. Overall, the experiments demonstrate that long-tailed object detection is primarily affected by object classification. Consequently, introducing rich semantics within classification data effectively alleviates this limitation. It is worth noting that this showcases our potential for further extension to unlabeled data.

| Method | AP | $AP_r$ |
|---|---|---|
| baseline | 32.2 | 24.1 |
| CLIP init backbone | 30.6↓1.6 | 22.2↓1.9 |
| RichSem ‡ | 35.0↑2.8 | 30.4↑6.3 |

(a) **Compared with CLIP initialized baseline.**

| Extra branch | AP | $AP_r$ | $AP_c$ | $AP_f$ |
|---|---|---|---|---|
| baseline | 32.2 | 24.1 | 29.9 | 38.3 |
| ✗ | 31.1↓1.1 | 24.6↑0.5 | 28.3↓1.6 | 37.2↓1.1 |
| ✓ | 35.0↑2.8 | 30.4↑6.3 | 33.1↑3.2 | 39.0↑0.7 |

(b) **Ablations on the semantic branch.**

| $\mathcal{D}^{extra}$ | AP | $AP_r$ |
|---|---|---|
| None | 32.2 | 24.1 |
| O365-Box | 33.0↑0.8 | 24.8↑0.7 |
| CC3M-Unl | 34.0↑1.8 | 28.7↑4.6 |
| INet-Unl | 34.7↑2.5 | 28.6↑4.5 |
| INet-LVIS | 35.0↑2.8 | 30.4↑6.3 |

(c) **Different extra datasets.**

| CLIP models | AP | $AP_r$ |
|---|---|---|
| None | 32.2 | 24.1 |
| RN50 | 35.0↑2.8 | 30.4↑6.3 |
| RN50×4 | 36.0↑3.8 | 33.0↑8.9 |
| RN50×16 | 36.2↑4.0 | 31.9↑7.8 |

(d) **Soft semantics provided by different CLIP models.**

| Extra data | $L_{soft}$ | AP | $AP_r$ |
|---|---|---|---|
| None | | 32.2 | 24.1 |
| INET-LVIS | ✗ | 33.8↑1.6 | 26.9↑2.8 |
| INET-LVIS | ✓ | 35.0↑2.8 | 30.4↑6.3 |
| INET-Unl | ✗ | 32.9↑0.7 | 23.7↓0.4 |
| INET-Unl | ✓ | 34.7↑2.5 | 28.6↑4.5 |

(e) **Soft semantics loss $L_{soft}$.**

| $\mathcal{D}^{backbone}$ | $\mathcal{D}^{od}$ | AP | $AP_r$ | $AP_c$ | $AP_f$ |
|---|---|---|---|---|---|
| IN-1K | LVIS | 32.2 | 24.1 | 29.9 | 38.3 |
| IN-21K | LVIS | 35.7 | 25.9 | 35.0 | 40.7 |
| IN-1K | LVIS+IN-21K | 35.0 | 30.4 | 33.1 | 39.0 |
| IN-21K | LVIS+IN-21K | 37.5 | 32.4 | 36.0 | 41.5 |

(f) **Ablations on backbone pre-training data and downstream detection data.**

| Method | AP | $AP_r$ | $AP_c$ | $AP_f$ |
|---|---|---|---|---|
| w/o $\mathcal{D}^{extra}$ | 32.2 | 24.1 | 29.9 | 38.3 |
| $+L_{soft}$ | 33.6 | 28.6↑4.5 | 32.4 | 37.2 |
| $+L_{loc}$ | 35.0 | 30.4↑1.8 | 33.1 | 39.0 |

(g) $L_{soft}$ **and** $L_{loc}$ **on extra data.**

| $\lambda_{soft}$ | AP | $AP_r$ | $AP_c$ | $AP_f$ | AP | $AP_r$ | $AP_c$ | $AP_f$ |
|---|---|---|---|---|---|---|---|---|
| | | # Only on LVIS | | | | # with Image-LVIS | | |
| 0 | 32.2 | 24.1 | 29.9 | 38.3 | 33.8 | 26.9↑2.8 | 31.5↑1.6 | 39.4↑1.1 |
| 0.2 | 32.4 | 23.8↓0.3 | 30.5↑0.6 | 38.5↑0.2 | 35.0 | 29.0↑4.9 | 33.3↑3.4 | 39.4↑1.1 |
| 0.5 | 32.4 | 25.0↑0.9 | 30.5↑0.6 | 37.7↓0.6 | 35.0 | 30.4↑6.3 | 33.1↑3.2 | 39.0↑0.7 |
| 1.0 | 32.1 | 27.0↑2.9 | 29.8↓0.1 | 36.9↓1.4 | 34.8 | 29.5↑5.4 | 33.4↑3.5 | 38.7↑0.4 |

(h) **Ablations on the weight of soft semantics loss.**

Table 3: **RichSem ablations.** All ablations are performed with RN50 as backbone under 1× schedule.

**Stronger semantics lead to better detectors.** We study the impact of CLIP with different backbones. The results are shown in Table 3d. With the capability increased, the model achieves better performance on overall AP. Besides, built upon a RN50×16, our detector obtains significant improvements on all metrics.

**Semantic learning is effective to leverage extra data.** Due to the data overlap between backbone pre-training and detector training, we conduct an ablation on them to further demonstrate the effectiveness of our method. As shown in the Table 3f, pre-training on large-scale data (ImageNet-21k) can provide strong perception capability for the downstream detection task, with overall performance gain. However, the performance on rare categories is still relatively low, indicating that this approach does not well alleviate the long-tail effects in detection. In contrast, our method is more effective than pre-training to handle long-tailed detection, especially for the tail categories. Notably, our approach is still effective with strong pre-trained backbones, further improving performance on long-tailed object detection.

**Semantics as soft targets leads to balanced learning.** The effectiveness of soft labels for low-shot categories can be reaffirmed in Table 3e: when using extra data, detectors trained with the soft semantics learning loss achieve better performance than those trained without the loss, especially for rare categories and unlabeled extra data. Besides, our semantic learning on extra classification boosts detection and classification at the same time thanks to our unified objective functions ( Section 3.3). As shown in Table 3g, both rich semantics and coarse locations play significant roles in boosting long-tail object detection. We further study the impact of soft classification objectives. We first experiment without using extra data. As shown in the Table 3h, with distillation weight set to 1, the performance of rare categories increases by ∼3AP, while the results of frequent categories decrease by 1.4. This suggests that semantics learning benefits those low-shot categories rather than overfitting frequent categories. As the distillation weight decreases, the performance of rare classes also drops but the results of frequent categories improve. After exploring additional data, the performance is no longer sensitive to the distillation weight. Thanks to the additional data, semantics learning leads to

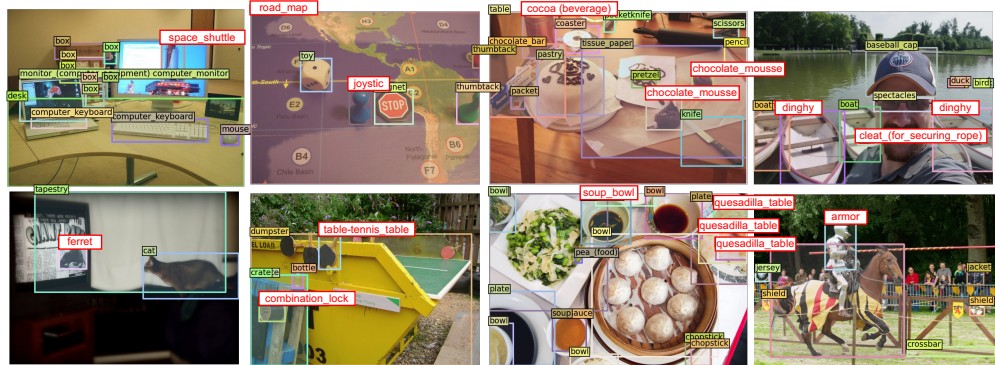

Figure 3: **Qualitative results of our RichSem.** We show the images containing rare categories from LVIS val 1.0. We show rare categories in red and show others in black. Best viewed on screen.

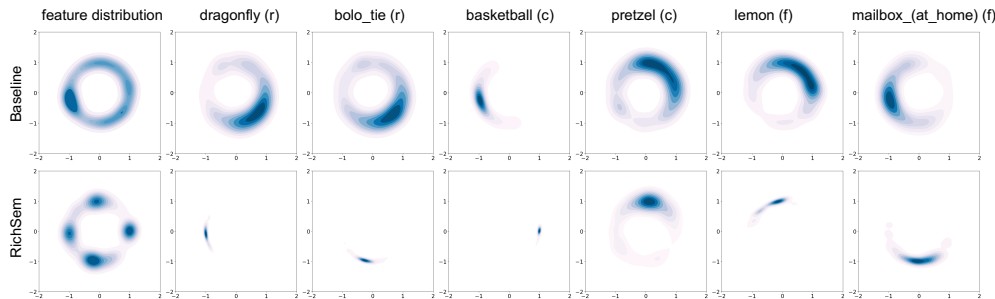

Figure 4: **Feature visualization between baseline and RichSem.** We randomly sample two classes each from the categories of rare, common, and frequent for visualization. Our RichSem shows well-clustered result across different categories.

consistent performance gains for all categories. We further visualize the object features of our method. Specifically, we normalize the features and employ Gaussian Kernel Density Estimation (KDE) in $\mathbb{R}^2$, following [52] to compare the distribution of object features across categories. As shown in the Figure 4, the visualization indeed shows a clear distinction between the baseline and our RichSem. Regarding the baseline, the distribution of object features lacks differentiation, often resulting in overlapping patterns among categories, especially between rare and frequent categories. In contrast, in RichSem, features belonging to each category, even rare categories, are well-clustered. This clear intra-class and inter-class distribution indicate that our approach effectively enhances the region classification capability of diverse categories. Therefore, our method effectively leads balanced learning among classes with varying frequencies.

## 5   Conclusions and Future Work

We presented RichSem, a simple but effective way, to leverage extra data by learning *rich semantics* and *coarse locations* to boost long-tailed object detection, alleviating the *semantics insufficiency* and *location sensitivity* caused by taking image-level labels as supervision. Through extensive experiments, we demonstrate our approach achieves state-of-the-art performance, without requiring complex training and testing procedures. One possible limitation is that we treat the detection data and classification data equally and use the same unified classification loss, which may be sub-optimal for the categories with sufficient samples.

For future work, we believe our framework can be extended to semi-supervised object detection (SSOD) where golden labels are not annotated on the unlabeled data, since our method is found to be robust when annotations are partially given. Moreover, the soft semantics learning for classification data can be naturally applied for open-vocabulary and multi-dataset detection tasks.

**Acknowledgement** This project was supported by National Key R&D Program of China (No. 2021ZD0112805) and National Natural Science Foundation of China (No. 62102092).

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
