# Learning from Rich Semantics and Coarse Locations for Long-tailed Object Detection

Lingchen Meng[1,2]   Xiyang Dai[3]   Jianwei Yang[3]   Dongdong Chen[3]   Yinpeng Chen[3]
Mengchen Liu[3]   Yi-Ling Chen[3]   Zuxuan Wu[1,2†]   Lu Yuan[3]   Yu-Gang Jiang[1,2]

[1]Shanghai Key Lab of Intell. Info. Processing, School of CS, Fudan University
[2]Shanghai Collaborative Innovation Center of Intelligent Visual Computing
[3]Microsoft

## A    Comparisons with large-scale vision foundation models.

Recently, advanced large-scale foundation models [5, 14] have shown exciting performance on downstream tasks. We use a huge size model Focal-H [15] and follow the trick utilizing the detection pre-training on Object365 used in [5, 14]. As shown in Table 1, our model achieves comparable performance with only 0.8B fewer parameters. Notably, our best model achieves balanced performance for both overall and rare categories. Although the perceptron capability of our backbone is weaker than other large-scale backbones, our 61.2 $AP_r$ outperforms EVA's 55.1 $AP_r$ by a large margin. This indicates that our method effectively boosts the detection capability of tail categories. Moreover, we did not use other training tricks [5, 14], *e.g.* enlarging the image size to $1.5\times$ when fine-tuning, soft NMS [2] or adopting test-time augmentations (TTA).

| Method | Detector | $\mathcal{D}_{det}$ | Backbone | Params | $\mathcal{D}_{backbone}$ | AP | $AP_r$ |
|---|---|---|---|---|---|---|---|
| ViTDet [10] | CMask R-CNN [3] | None | ViT-H-MAE [7] | 692M | IN-1K | 53.4 | n/a |
| EVA [5] | CMask R-CNN [3] | O365 | EVA-H [5] | 1.1B | merged-30M[a] | 62.2 | 55.1 |
| InternImages [14] | DINO [16] | O365 | DCNv3-H [14] | 2.2B | merged data[b] | **63.2** | n/a |
| Ours | DINO [16] | O365 | Focal-H [15] | 747M | IN-22k | 61.2 | **61.2** |

Table 1: **Comparison with SoTA on LVIS val 1.0.** $\mathcal{D}_{det}$ indicates the datasets used in detector pre-training. $\mathcal{D}_{backbone}$ indicates the datasets used in backbone pre-training. "n/a" indicates the numbers are not available for us. 'merged-30M[a]': IN-21K + O365 + COCO + ADE20K + CC15M. 'merged data[b]': Laion-400M + YFCC-15M + CC12M

## B    Robustness Analysis

**Soft-labels are important for improved balanced object classification (Rich semantics).** We use CLIP to perform zero-shot object classification on LVIS val 1.0. We obtain CLIP object features according to their ground truth bounding boxes and classify them using the contrast with textual features of categories. In addition, we use classification accuracy to reflect the quality of semantics from CLIP. We see from Table 2 that although the Top-1 accuracy is relatively low, the Top-10 accuracy is around 34%, indicating that CLIP can properly rank labels into the top classes rather than hard classification. Most importantly, it is encouraging to see CLIP has a balanced performance among rare, common, and frequent categories. In light of the above points, using soft labels for object semantics from CLIP can provide "good guidance" for the balanced performance.

**Soft-labels provide better robustness towards location shifts (Coarse locations).** We further study the robustness towards location shifts by adding noise to ground truth boxes. As shown

in Figure 1, when the noise scale is relatively small ([0, 0.5]), the top-10 performance only drops slightly, suggesting that soft labels are robust to the quality of bounding boxes for classification: when bounding boxes shift slightly, these semantics change slowly. Moreover, when the noise scale is large (>0.5), the performance drops drastically due to the inaccurate boxes, on which the ground truth class labels mismatch the semantics of the noised boxes. Similarly, different crops may lead to the mismatching between cropped semantics and ground truth labels. In contrast, semantics derived from soft labels can well represent the semantics within the crop, for it is adaptive to locations or crops.

| Object classification | AP | $AP_r$ | $AP_c$ | $AP_f$ |
|---|---|---|---|---|
| Top1 class per proposal | 16.2 | 16.7 | 16.4 | 15.7 |
| Top5 classes per porposal | 29.6 | 29.9 | 29.3 | 29.8 |
| Top10 classes per porposal | 33.9 | 33.1 | 33.7 | 34.6 |

Table 2: **CLIP zero-shot object classification on LVIS val v1.0.** We use pre-trained CLIP-RN50 [11] to extract region features according to the ground truth bounding boxes and perform object classification. CLIP achieves a balanced performance among *rare*, *common* and *frequent* categories.

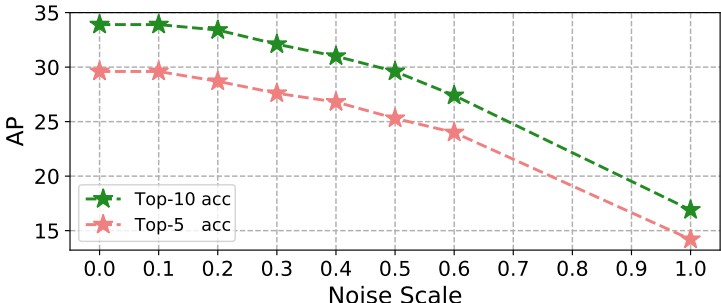

Figure 1: **CLIP zero-shot object classification under different noise scales.** We use Top-5 and Top-10 object classification accuracy to reflect the CLIP semantics robustness towards location shifts.

**Semantics learning leads to the robustness of annotations.** We also find that our semantics learning scheme is robust to annotations. To verify this, we randomly drop a part of ground truth annotations on the target detection dataset while ensuring each category has at least one training sample. We first show the robustness of the detection dataset. As shown in Figure 2 (a), when trained on LVIS only, the performance with and without semantics learning is close when all annotations are used. When the dropping ratio increases, the significance of semantics learning is more clear. We also show in Figure 2 (b) that with the help of the rich semantics in INet-LVIS, our detector can achieve even better performance with just 50% annotations.

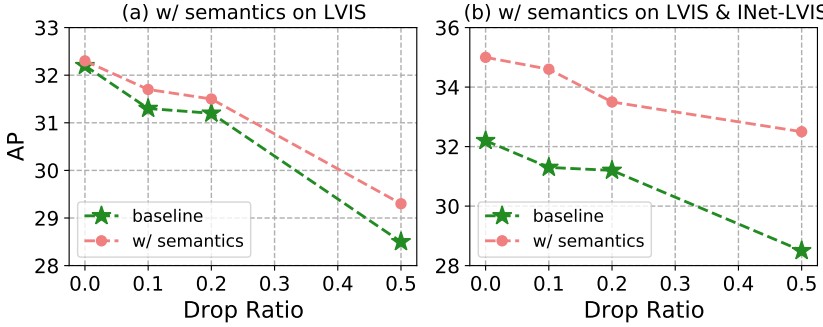

Figure 2: **Ablation on robustness of partial annotations.** We random drop a part of ground truth accoridng to drop ratio.

## C Datasets

**Detection data.** We conduct our experiments on three datasets of long-tailed distribution: LVIS [6], OpenImages [9], and Visual-Genome [8]. We mainly evaluate our method on LVIS with the official LVIS evaluator. We evaluate OpenImages and Visual-Genome under a COCO-style evaluator; we report AP for OpenImages, and AP, $AP_{50}$ for Visual-Genome, respectively. We only compare our method with our baseline model on Visual-Genome and OpenImages since the datasets are not popular long-tail object detection benchmark datasets, which is hard to find the previous method's performance for a fair comparison.

**Extra data.** We experiment on three extra data: Object365 [12], ImageNet-21k [4] and CC3M [13]. Object365 dataset contains around 0.6M images with 365 classes. Each image is densely annotated by human labelers to ensure quality. ImageNet-21k is for classification, containing 14M images with 21K image-level category labels. CC3M contains 3M image-text pairs from the web. We summarize the datasets used in our experiments below.

| Notation | Imgs | Annotation | Definition |
|----------|------|------------|------------|
| LVIS | 0.1M | bounding boxes and classes | The original LVIS [6] |
| O365 | 0.6M | bounding boxes and classes | The original Object365 [12] |
| INet-21k | 14M | image-level class label | The original ImageNet-21k [4] |
| CC3M | 3M | image-level description | The original CC3M [13] |
| INet-Unl | 14M | no annotations | INet-21k w/o labels |
| INet-LVIS | 1M | image-level label | INet-21k classes overlapped with LVIS |
| O365-Box | 0.6M | bounding boxes | O365 w/o class labels |
| CC-Unl | 3M | no annotations | CC3M w/o labels |

**Long-tailed Frequency Analysis** We visualize the number of instances for each category in LVIS [6], OpenImages [9], and Visual-Genome [8]. As shown in Figure 3, categories in three datasets all follow long-tailed distributions, *i.e.* the number of instances on the head class is $\sim 10^3$ times than those on the tail. Moreover, the instance number of rare categories in LVIS is extremely low (less than 10), which makes training challenging. However, RichSem achieves significant improvements on rare categories with the help of image classification datasets, suggesting that RichSem can improve the capability under the low-shot learning setting.

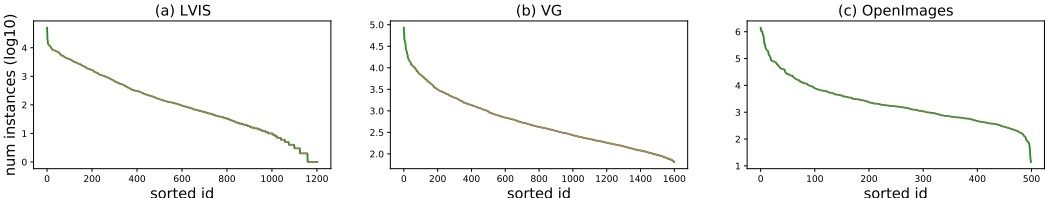

Figure 3: **The number of instances per categories.**

## D Additional Ablations

### D.1 Results on other long-tailed datasets.

We conduct experiments on the other two datasets of long-tail distribution, OpenImages [9] and Visual-Genome [8]. We use INet-Unl as extra data and train as Eq.7. This allows us to use ImageNet-21k directly without manual label mapping. We use ResNet50 as backbone and train all models under $2\times$ schedule. As shown in Table 3, RichSem obtains 0.5 and 2.1 AP gain compared with the baseline. The exciting performance gains on both Visual-Genome and OpenImages demonstrate that our proposed method is effective to those dataset of long-tailed distribution.

| RichSem | Visual-Genome | | OpenImages |
|---|---|---|---|
| | AP | AP$_{50}$ | AP |
| ✗ | 7.3 | 11.9 | 38.9 |
| ✓ | 7.8↑0.5 | 12.3 | 41.0↑2.1 |

Table 3: **Experiments on Visual-Genome and OpenImages.** We use ResNet50 as the backbone for all experiments; ✗ indicates that the baseline that only trained Visual-Genome and ✓ indicates that the model is trained with INet-Unl using RichSem.

## D.2 Rigorous Comparison with Detic

To further claim the effectiveness of our method compared with Detic, we design the following two ablation experiments:

**Detic → Our baseline**: Since we choose DINO as our baseline detector, we reimplement Detic in our baseline.
**Ours → Detic baseline**: We also integrate our RCLT into Detic baseline detector [18] (CenterNet2-CasccadeRCNN).

Besides, we keep the other experiment settings exactly the same. As shown in Table 4, our method outperforms Detic under both baselines, especially AP of rare categories.

| Detector | Method | AP | AP$_r$ |
|---|---|---|---|
| CenterNet2-CasscadeRCNN [18] | baseline | 31.5 | 25.6 |
| CenterNet2-CasscadeRCNN [18] | Detic [17] | 33.2 | 29.7 |
| CenterNet2-CasscadeRCNN [18] | RichSem (Ours) | **33.5** | **31.0** |
| DINO [16] | baseline | 32.2 | 24.1 |
| DINO [16] | Detic [17] | 33.8 | 26.9 |
| DINO [16] | RichSem (Ours) | **35.0** | **30.4** |

Table 4: **Rigorous Comparison with Detic.**

# E   More Implementation Details

## E.1 Semantics Learning

We use a RN50 to extract object semantics for most experiments, unless metioned otherwise. We freeze all parameters in CLIP. As for the proposed semantic branch, we only use a Linear layer to project the object feature into the semantics space and perform CONTRAST with the text features of categories to obtain the semantics prediction. To achieve a balanced performance, we set the weight of soft loss $\lambda_{soft}$=0.5 for all main results.

## E.2 Data augmentations

Following DINO [16], we use a standard training augmentation for all experiments. We randomly resize an image from the original detection dataset with a shorter edge between 480 and 800 and limit its longer edge below 1333. For extra data, we random crop and use mosaic [1] to provide coarse location positions.

# F   More Visualizations

We further provide more qualitative results in addition to those in the main text. Moreover, we compare them with those qualitative results predicted by the detector trained without INet-LVIS as extra data. As shown in Figure 4, RichSem can better detector rare categories, such as the "leather" and "gas mask" in the first two columns, which is ignored by the baseline detector.

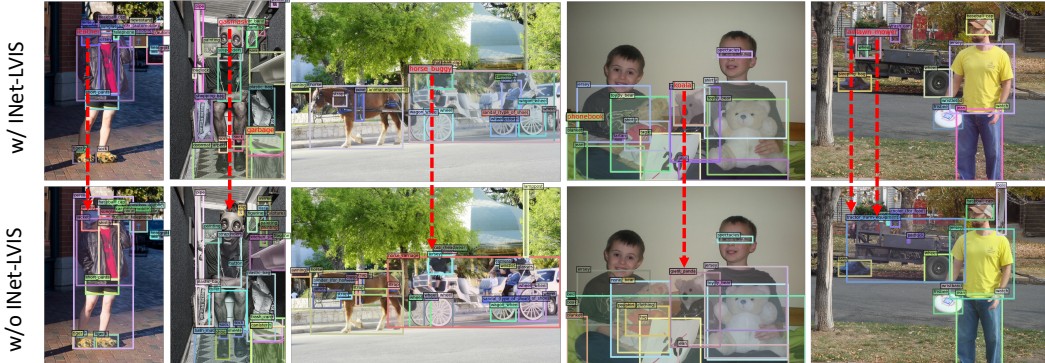

Figure 4: **Qualitative results of our RichSem.** We visualize the prediction of RichSem with INet-LVIS as extra data and compare them with our baseline model without extra data. We show rare categories in red and show others in black. RichSem with extra data can learn better on rare categories. Red arrows mean those rare objects detected by RichSem while not detected in our baseline model.