# OpenReview forum: "Learning from Rich Semantics and Coarse Locations for Long-tailed Object Detection"
_NeurIPS.cc/2023/Conference — NeurIPS 2023 poster_

### Official Review · Reviewer_vo8U · 2023-06-16

**Soundness:** 2 fair
**Presentation:** 3 good
**Contribution:** 3 good
**Rating:** 5
**Confidence:** 4

**Summary:**

This paper uses the vision-language pre-trained model CLIP for long-tailed object detection.
The key idea is to consider not only image-level semantics but also region-level semantics, and fuse them under a soft-label scenario.
The overall idea is deployed on different object detectors with different backbones.
Experiments are conducted on three long-tailed object detection datasets with extra data.
Besides, the experiments and discussion on ablation studies and foundational models are very extensive.


**Strengths:**

+ To the best of the reviewer’s knowledge, it is the first work (excluding CVPR2023) to use vision-language pipelines for long-tailed object detection. The contribution is significant.

+ The ablation studies and other discussions are very extensive, in both the main submission and the

+ This paper is well-written and easy-to-follow.


**Weaknesses:**

- This work, in its current form, lacks theoretical insight, especially on how the proposed method fits into the long-tailed feature distribution.

As the proposed method is targeted at the context of long-tailed object detection, not the generic object detection, it is very necessary to extensively discuss how the framework and the specific module design can work well or fit into the long-tailed feature distribution.

Unfortunately, as the reviewer goes through the entire manuscript: In Fig.1 motivation case and Fig.2 framework, there is no element for the long-tailed feature space or long-tailed elements. Besides, in the methodology section, from learning the image / object semantics to use soft labels, all these presentations are generic and are suitable for generic recognition, detection, segmentation and etc. More importantly, in the experimental section, there is no feature space visualization on how CLIP benefit the long-tailed feature distribution.

- The state-of-the-art comparison is insufficient, and to some extent, unfair. Specially,

It is really strange that in Table 1, the authors only compare some generic vision transformer based detectors such as DERT. Why the methods, using vision-language pre-trained model for generic object detection, such as [11,18,27,34,57], are not involved for comparison on the long-tailed object detection datasets? It is very likely that these methods [11,18,27,34,57] for generic vision-language object detection can work well on long-tailed detection datasets.

- The performance of the proposed method seems to be mainly brought high by the use of Swin-Transformer.

This remark is not saying that the state-of-the-art performance with Swin-Transformer backbone is not important, but the reviewer would like to raise the attention that, when use the old ResNet-50 backbone (Row 1 of Table 1), the proposed method, even with vision-language pre-trained model, does not lead a significant performance against some SOTA in 2021.

Thus, the reviewer feels that in the future, why with Resnet-50 is not that effective for the proposed framework, worth to be further investigated.

Other minor issues to improve:

- Extensive feature visualization on the context of both long-tail and CLIP features, please.

- Please use \mathcal{} to present the loss function, and use \mathbf{} to present the tensors and vectors. These notations can distinguish them from scalars.


**Questions:**

The reviewer really appreciate that the work is the first to extend vision-language pre-trained model to the long-tailed object detection task. However, the weakness of this submission is also very obvious.

In the rebuttal, please address the weakness part point-by-point, on lack insight for long-tailed context, lack state-of the-art comparison, and to-some-extent performance ineffectiveness.


**Limitations:**

The limitation is not properly discussed

---

> ### Author Rebuttal · Authors · 2023-08-10
>
> ### q1: Insight for long-tail context.
> Thanks for pointing out this. Exploring extra data is indeed a direct and effective approach to mitigating data scarcity. Our primary motivation arises from the fact that classification data is easy to collect and offers a more balanced distribution compared to detection data.
>
> For instance, consider the ImageNet-21k and LVIS; there are 997 overlapped categories in ImageNet-21k, covering a significant portion of the LVIS categories with a 10$\times$ number of images, greatly enhancing instance diversity. Furthermore, the samples in ImageNet-21k exhibit a balanced distribution (please refer to Fig.1 in our **rebuttal pdf**). Therefore, leveraging classification datasets demonstrates great potential in alleviating data scarcity.
>
> However, it is still challenging, summarized as (1) semantic ambiguity and (2) location sensitivity. Previous works mainly focus on box estimation to solve location sensitivity but neglect semantic ambiguity and the importance of rich semantics inside classification data.
>
> Fortunately, V-L pre-trained models (CLIP) have demonstrated powerful zero-shot recognition capabilities, benefiting from web-scale training data. Our pilot experiments (please refer to Tab.1 and Fig.2 in the **rebuttal pdf**, and Sec. B in supplementary) illustrate that CLIP models exhibit balanced performance across rare, common, and frequent categories; and show robustness towards location shift.
>
> Given these compelling factors, it becomes natural to utilize CLIP models and delve deeply into leveraging the rich semantics from classification data to address long-tail object detection effectively.
>
>
> ### q2: Lack of state-of-the-art comparison
>
> Current research on vision-language (V-L) pretraining and foundational models for long-tailed object detection can be categorized into two main directions:
>
> #### 1) **Leveraging the well-aligned Vision-Language knowledge of the pre-trained**
> These works have primarily focused on open-vocabulary detection tasks.
>
> For a fair comparison, we thus conduct an experiment on open-vocabulary LVIS, which can be viewed as an extremely long-tailed distribution where tail categories have zero occurrences.
>
> * Open Vocabulary LVIS detection results compared with [11,18,27,34,57]
>
> | Method | backbone |AP | AP_novel
> |---- | ---- |---- |---- |
> | CLIP [34] + gt bbox| ViT-B| 17.7 |18.9 | 18.8 | 16.0
> | GLIP-zeroshot [27]| Swin-L| 26.9 | 17.1 | 23.3 | 35.4 |
> | ViLD [11] | R50|  27.5 |17.4 | - | - |
> | RegionCLIP [57]| R50| 27.4 | 17.0 | 26.7 | 32.9 |
> | Ours| R50 | 31.5 | **23.0** | 30.9 | 36.0
>
> As shown in the table, our method surpasses the mentioned previous sota on their benchmark with nearly 6 AP gains on novel categories, further demonstrating the effectiveness of our method.
>
>
> #### 2) **Scaling up the pretraining data and model size**
>
> Researchers employ large vision foundation models and advanced training techniques to achieve excellent performance, regardless of the computational burden and training recipe consistencies. In contrast, we utilize the advanced Focal-H to compare with these works on LVIS detection.
>
> * Results on LVIS val v1.0
>
> | Method | backbone | $D_{backbone}$ | param|AP | AP_r | AP_c | AP_f|
> |---- | ---- |---- |---- |---- |---- |---- |---- |
> | ViTDet| ViT-H-MAE | IN-1k | 692M | 53.4 | - |- |
> | EVA | EVA-H | merged-30M | 1.1B | 62.2 | 55.1| 62.2| 65.2 |
> | InternImage| DCNv3-H | merged-data | 2.2B |63.2 |- |- | - |
> | Ours| Focal-H | IN-22k | 747M | 61.2 | **61.2** | 60.1 | 62.4
>
> The table shows our model achieves comparable performance with 0.8B fewer parameters, offering a pretty balanced performance for both overall and rare categories.
>
> Notably, we did not use other advanced training tricks, e.g., large image size (1.5$\times$ larger) and batch size training (16 vs. 64), and their additional inference post processings (soft NMS and TTA).
>
> ### q3: Performance "ineffectiveness" on ResNet-50
> Sorry for the confusion. In fact, our method is indeed **effective** on R50.
>
> Two main reasons are contributing to the "ineffectiveness" :
>
> 1. Backbone pretraining.
> The previous sota (Detic) utilized R50-21k [1] (refer to R50$\star$ in Tab.1 of the main paper), offering strong perception capability for downstream detection. In contrast, we just use the R50-1k, which is significantly weaker than R50-21k.
> 2. The size of the CLIP model providing semantics.
> We utilize CLIP-RN50 as the default semantics provider. However, the previous sota uses CLIP-ViT-B to generate the weight of the classifier. We observe that stronger semantics lead to better performance (L297-L300 in our main paper), indicating that our method can be further improved.
>
> Based on these, we conducted experiments with the stronger backbone and semantics provider. Our best model surpasses the previous sota by a large margin (4.1 AP and 6.1 AP_r), indicating its effectiveness on the R50 backbone.
>
> | Method  | Backbone | CLIP model  |AP | AP_r | AP_c | AP_f|
> | ---- |---- |---- | ---- |---- |---- |---- |
> | Detic  | R50-21k  | CLIP-Vit-B | 36.8 | 31.4 | 36.0 | 40.1
> | Ours  | R50-1k  | CLIP-RN50 | 37.1 | 29.9 | 35.6 | 42.0
> | Ours  | **R50-21k** | CLIP-RN50| 40.1 |36.2 | 38.2 | 44.0
> | Ours  | **R50-21k** | **CLIP-RN50x4** | 40.9 | 37.5 | 39.6 | 43.8
>
>
> Thanks for pointing it out and for the great inspiration; we will update this table in the next version.
>
> [1] Imagenet-21k pretraining for the masses. In NeurIPS, 2021.
>
>
> ### q4: Limitation part.
>
> A main limitation of our approach is treating detection data and classification data statically with strict equality in the unified objective. An optimal scenario might dynamically entail prioritizing $L_{cls}$ for instances that can be accurately categorized while making $L_{soft}$ precedence in other situations.
>
> ### q5: Suggestions on fonts of notations and feature visualization
>
> Thanks for your suggestions, and we will refine the fonts to make them distinguished and visualize the features in the next version.

---

> > ### Comment · Reviewer_vo8U · 2023-08-10
> > **Response to author rebuttal by Reviewer vo8U**
> >
> > Thanks for the authors to provide such a detailed rebuttal.
> >
> > Indeed, my concerns on Question 1, 2, 3 and 4 have been well addressed, which I appreciate a lot.
> >
> > However, before the discussion end period, could the authors also provide how the proposed pipeline can improve the feature space of long-tailed tasks against the baseline?
> >
> > I feel this is necessary for the scope of NeurIPS, and it can convince me to raise my score above accept threshold.
> >
> > Many thanks, and look forward to the update !

---

> > > ### Author Response · Authors · 2023-08-13
> > >
> > > Thank you for your timely response and valuable suggestions.
> > >
> > > We have performed a visualization of object features to effectively demonstrate the advantages of our approach within the feature space. Although we had planned to upload the visualization to provide a deeper insight, unfortunately, this year's guidelines prohibit external links during the discussion period. After consulting with the ACs, they suggested that "authors can promise and describe the visualizations they are planning to add and what they show in words." Following the guideline, we provide a detailed description of our visualization pipeline and the resulting outcomes below, and we apologize for the inconvenience.
> > >
> > > Firstly, we extract **object features** from the validation set corresponding to their ground truth bounding boxes. These extracted features are then projected into a 2D space using PCA. For a clear visualization, we randomly select six categories, encompassing two rare, two common, and two frequent categories.
> > >
> > > To visualize the distribution, we normalize the features and employ Gaussian Kernel Density Estimation (KDE) in $\mathbb{R}^2$, following [1]. This visualization technique offers us a way to compare the distribution of object features across categories. Furthermore, it provides a comparative analysis between the baseline and our proposed RichSem.
> > >
> > > As a result, we have a visualization similar to Fig.3 in [1]. The visualization indeed shows a clear distinction between the two models. Regarding the baseline, the distribution of object features lacks differentiation, often resulting in overlapping patterns among categories, especially between rare and frequent categories. In contrast, in RichSem, features belonging to each category, even rare categories, are well-clustered. This clear intra-class and inter-class distribution indicate that our approach effectively enhances the region classification capability of diverse categories.
> > >
> > > These observations highlight the effectiveness of our proposed method in long-tail object detection, particularly in improving the performance of rare categories.
> > >
> > > We greatly appreciate your valuable insights and suggestions. We promise to include the visualization in our next version.
> > >
> > > [1] Wang, Tongzhou, and Phillip Isola. "Understanding contrastive representation learning through alignment and uniformity on the hypersphere." ICML, 2020.

---

> > > > ### Comment · Reviewer_vo8U · 2023-08-13
> > > > **Response to author rebuttal (Round 2)**
> > > >
> > > > Thanks for the authors to further address my rest concerns.
> > > >
> > > > In this regard, I am willing to improve my rating to above the accept threshold, as long as the authors can provide visualized feature space in their camera-ready version.
> > > >
> > > > Kind regards.

---

### Official Review · Reviewer_gFff · 2023-07-04

**Soundness:** 4 excellent
**Presentation:** 4 excellent
**Contribution:** 3 good
**Rating:** 7
**Confidence:** 5

**Summary:**

- This work deals with Long-tail object detection. Authors identify two problems with using additional data, namely Semantic ambiguity and Location sensitivity.
- Authors identify that semantic ambiguity arises due to supervision with one-hot encoded labels from the image datasets and instead propose to use CLIP scores for supervision.
- CLIP's ability to provide sufficient semantic information even with course locations is leveraged to tackle location sensitivity.
- Authors propose RichSem, a simple yet effective method, that adds an additional semantic branch to the detector to learn rich semantics from images.
- The semantic branch is only required during training and with extensive experiments is shown to achieve state-of-the-art results on LVIS dataset in overall and rare categories.

**Strengths:**

- The paper is well written and the presentation makes it easy to follow.
- The experiment section is exhaustive and supports all the claims made by the authors. Authors test their method with the transformer and R-CNN based family of detectors and the ablation experiments clearly explain the contribution of each component.
- RichSem is a principled way to leverage additional data for long-tailed object detection.

**Weaknesses:**

- Authors miss the comparison with [1], which also uses additional data. Please compare appropriately in Table-1.
- The current work heavily relies on CLIP but it is widely known that CLIP has several limitations [2]. It would be interesting to address the robustness of the current method to the limitations of CLIP.

[1] Bo Li, Yongqiang Yao, Jingru Tan, Xin Lu, Fengwei Yu, Ye Luo, and Jianwei Lu, Improving Long-tailed Object Detection with Image-Level Supervision by Multi-Task Collaborative Learning.

[2] https://stanislavfort.github.io/blog/OpenAI_CLIP_stickers_and_adversarial_examples/

**Questions:**

- Authors mention constructing a mosaic with multiple images in L51, where in the whole method is this done?
- What is $f_v^i$ in Eq. 4? Is that $s^t$ obtained in Eq. 2?
- From table 2c, rows 4,5 the increase in improvement is because of image level labels. But how are the annotations used in the whole pipeline? The semantic branch only uses CLIP similarities for the KL divergence loss? Is a hard supervised loss also being applied for the image labels? Is there a localization term for the image level labels?
- What is $f^t$ in the semantic branch in Fig. 2? If $t\in (\text{loc},\text{cls})$, then is the semantic soft loss applied to both $f^{\text{loc}}$ and $f^{\text{cls}}$?

**Limitations:**

I do not foresee any potential negative societal impact of this work.

---

> ### Author Rebuttal · Authors · 2023-08-10
>
> ### q1: Comparison with [1]
> Thanks for the good suggestion. We incorporate our method into Faster-RCNN, employing R50 as the backbone for an appropriate comparison following [1].
>
> The table shows that our method achieves a strong performance and surpasses the previous sota by more than 3 AP on rare categories. Unlike CLIS[1], our approach focuses on digging rich semantics with only pre-defined bounding boxes on extra classification data. Thanks to our semantic learning scheme, our model can effectively leverage the information within classification data for long-tail object detection, achieving a more balanced performance between head and tail categories.
>
> We greatly appreciate your suggestion and will update the table in our next version.
>
> | Method  |AP | AP_r | AP_c | AP_f|
> |---- | ---- |---- |---- |---- |
> | Faster RCNN| 24.1 | 14.7 | 22.2 | 30.5
> | EQLv2| 25.5 | 16.4 | 23.9 | 31.2 |
> | BAGS| 26.0 | 17.2 | 24.9 | 31.1
> | Seesaw| 26.4 | 17.5 | 25.3 | 31.5
> | EFL| 27.5 | 20.2 | 26.1 | 32.4
> | MosaicOS| 23.9 | 15.5 | 22.4 | 29.3
> | CLIS| 29.2 | 24.4 | 28.6 | 31.9
> | Ours | 30.6 | **27.6** | 29.7 | 32.9
>
> [1] Improving Long-tailed Object Detection with Image-Level Supervision by Multi-Task Collaborative Learning.
>
> ### q2: Robustness of CLIP
> Thank you for your suggestion. We do agree CLIP is not robust towards adversarial attacks like many other types of neural networks. In this work, we assume images are pristine, and we will study the robustness of CLIP in the future.
>
> ### q3: Questions on mosaic augmentation
> Thanks. We only apply 2x2 mosaic augmentation on the **extra classification data**.
> Specifically, we utilize a pre-defined whole-image bounding box for each image and use the mosaic augmentation [1] to randomly concatenate sub-images into a mosaic, thus offering coarse locations on classification data.
>
> Unlike previous works on extra data focusing on bounding box estimation, such as using a pre-trained detector as region generators [2], online predictions [3], and post-processing methods [4],  we emphasize that coarse locations are sufficient and pay more attention on introducing rich semantics from the classification data to the detector.
>
>
> [1] Yolov4: Optimal speed and accuracy of object detection
> [2] Improving Long-tailed Object Detection with Image-Level Supervision by Multi-Task Collaborative Learning
> [3] Detecting Twenty-thousand Classes using Image-level Supervision
> [4] MOSAICOS: A Simple and Effective Use of Object-Centric Images for Long-Tailed Object Detection
> ### q4: Eq.4
> Sorry for the confusion. In Equation 4, $f^t_{i}$ refers to the corresponding semantic guidance obtained from CLIP, denoted as $s^t$ earlier. In Eq. 4, we compute the Kullback-Leibler divergence between the soft semantic prediction and the corresponding semantics provided by CLIP models. For the sake of consistency, they should be $o^{soft}$ and $s^{t}$ instead of $f^t_{i}$ and $s^t$ in Eq.4. We will definitely refine the notations in the next version.
>
> ### q5: Questions on rows 4-5 of Tab 2c
> Both line 4 (ImageNet-Unl) and line 5 (Image-LVIS) incorporate the soft KL loss, hard classification loss, and location loss in our training pipeline.
>
> For the soft KL loss and location loss, both line 4 and line 5 follow the same approach. They utilize pre-defined whole-image bounding boxes as pseudo-locations and extract semantics based on these coarse locations, forming the soft targets of the semantic branch.
>
> Regarding the hard classification loss, line 5 employs image-level labels that are mapped to the LVIS taxonomy as the target. On the other hand, for line 4, the pseudo hard labels for classification are generated by using the class with the highest logits in each semantic target (as described in Eq. 7). Additionally, we incorporate a threshold $th$=0.05 to filter out images that are too irrelevant to the taxonomy.
>
> Due to the distinctions above, line 5 (Image-LVIS) slightly outperforms line 4 (Image-Unl). This improvement can be attributed to the well-matched taxonomy between the extra data and the target detection data, along with a well-designed label mapper.
>
> However, it is noteworthy that the difference between Image-LVIS and Image-Unl is relatively small (only 0.3 and 1.8 on AP and AP_rare). This observation shows our potential for unlabeled classification/object-centric data.
>
> ### q6: Questions on f^t
>
> $f^t$ in Fig.2 represents the object-level features from CLIP (please refer to Section 3.1 and Equation 2). More specifically, $f^t$ is derived by pooling CLIP features based on the corresponding bounding box. For detection data, this bounding box corresponds to the tight ground-truth bounding box, while for classification data, it involves a coarse whole-image bounding box. The object-level CLIP feature $f^t$ is then employed to derive $s^t$ by calculating the Contrast with linguistic categories features $f^{cat}$, which is soft semantic guidance in our training scheme.

---

> > ### Comment · Reviewer_gFff · 2023-08-12
> >
> > I would like to thank the authors for their detailed rebuttal.
> > I'm satisfied with the author's response and would recommend adding these discussions to the final version. After looking at the other reviews and the author's rebuttals, I vote to accept this paper.

---

### Official Review · Reviewer_ndNj · 2023-07-06

**Soundness:** 3 good
**Presentation:** 3 good
**Contribution:** 2 fair
**Rating:** 6
**Confidence:** 3

**Summary:**

To address semantic ambiguity and location sensitivity, this paper introduces a one-stage training framework that leverages additional image data to boost the detector through learning from rich semantics and coarse locations for long-tailed object detection. And their RichSem achieves consistent improvements on both overall and rare-category of LVIS under different backbones and detectors.

**Strengths:**

1. The paper is clearly written. And main idea of the paper is easy to understand.
2. This paper introduces a novel semantics learning framework, which uses an additional branch to learn from rich semantics and coarse locations for long-tailed object detection without the need to compute pseudo labels.
3. Extensive method demonstrates significant results on long-tailed datasets

**Weaknesses:**

1. In Long-tailed object detection, the scarcity of samples or natural constraints results in a limited number of instances in the tail classes. Can exploring extra data effectively address the issue of scarce tail classes in practical applications?
2. CLIP is a large-scale model designed for the joint processing of images and text. The lack of representative samples for tail classes may lead to a relatively weak understanding and recognition ability of CLIP for these classes. Can CLIP still provide stable semantic guidance?

**Questions:**

Please check the paper weaknesses.

**Limitations:**

Please check the paper weaknesses.

---

> ### Author Rebuttal · Authors · 2023-08-10
>
> ### q1: Can exploring extra data effectively address data scarcity?
>
> Yes. Indeed, exploring additional data is a straightforward approach to enhancing the performance of tail categories. However, acquiring bounding box annotations for these rare categories is labor-intensive and costly.
>
> Recognizing this challenge, we introduce a new approach that leverages existing image classification data to alleviate the data scarcity issue faced by tail categories.
>
> Unlike conventional semi-supervised or weakly supervised methods, our approach does not require estimating bounding boxes, which is challenging. Instead, we focus on rich semantics obtained by CLIP models and leverage coarse locations from classification data for improved results.
>
> We conduct comprehensive experiments involving varying backbones, detectors, schedules, and datasets. Our method shows consistent gains in all experiments, demonstrating that our method can effectively address the challenge caused by the scarcity of detection data.
>
>
> ### q2: Can CLIP provide stable semantics on rare categories?
>
> Yes, CLIP models showcase balanced recognition capabilities across rare, common, and frequent categories thanks to the web-scale image-text pairs as training data (see supp sec.B, we also include the table here for convenience).
>
> We use CLIP-RN50 and perform region classification on LVIS utilizing ground truth bounding boxes. Specifically, we obtain object features according to their ground truth bounding boxes and classify them using the contrast with textual features of categories.
> As shown in the table below, the AP of the top 10 predictions per proposal is around 34%, indicating that CLIP can properly rank labels into the top classes. Furthermore, a key observation is that the results highlight a well-maintained and balanced performance across categories with varying frequencies.
>
>
> | Region classification  |AP | AP_r | AP_c | AP_f|
> |---- | ---- |---- |---- |---- |
> | top1 class per proposal| 16.2 | 16.7 | 16.4 | 15.7
> | top5 class per proposal| 29.6 | 29.9 | 29.3 | 29.8
> | top10 class per proposal | 33.9 | 33.1 | 33.7 | 34.6
>
> We also find that CLIP models are robust towards location shifts. As shown in Fig.2 in the **rebuttal pdf**, we gradually introduce noise to ground truth boxes, and the top-10 performance experiences only a marginal drop when the noise scale remains relatively small (ranging from 0 to 0.5).
>
> These observations underscore CLIP's capacity to provide stable and precise semantic information. It allows our method to effectively address data scarcity by leveraging extra classification data.

---

> > ### Comment · Reviewer_ndNj · 2023-08-16
> >
> > I would like to thank the authors for the clarification, which solves most of my concerns. But im my humble opinion, the performance of this method heavily relies on more extra data and the CLIP model, which limits the potential for widespread impact of the work. Can authors further discuss this issue？

---

> > > ### Author Response · Authors · 2023-08-16
> > > **Response to Reviewer ndNj**
> > >
> > > Thanks for your response and suggestions.
> > >
> > > Given the natural statistics of long-tail distribution, current detectors easily bias towards the head categories and show poor performance on the tail categories, limiting their broader applications. Our proposed method effectively addresses this challenge by capitalizing on extra classification data and leveraging the knowledge encoded in pre-trained vision-language models (VLMs), making the detectors more practical and broadly used.
> > >
> > > It's important to highlight that leveraging extra data [1,2,3], such as weakly annotated or unlabeled data, and the knowledge from pre-trained vision-language models (VLMs) [3,4,5] shows great potential in scenarios with limited training resources of tail categories. Compared to the previous works on improving the training recipe like loss re-weighting [6], data re-sampling [7], augmentation [8], and decouple training [9], it can directly address the data scarcity by increasing the amount and diversity of training instances.
> > >
> > > As for long-tail object detection, we can utilize CLIP models as semantics providers and classification data, like ImageNet-21k, as extra data, both of which are readily available. Compared to collecting and annotating detection directly, leveraging classification data is more efficient without additional data collection and annotation. Besides, the great generalization capability of VLMs makes us extract rich semantics without any fine-tuning. Overall, these allow for the straightforward application of our method to the detector, resulting in effectiveness and efficiency. Additionally, our proposed semantic branch helps the detector learn the soft semantics within extra data and enhance the feature representation during training, and it can be removed during inference, improving the method's flexibility for use with various detectors.
> > >
> > > Furthermore, our method can be extended to different types of extra data, including well-label-mapped classification data (INet-LVIS), unlabeled classification data (INet-Unl), and web-collected image-text pairs (CC3M-Unl). It's also adaptable to various sizes of CLIP models (CLIP-RN50, CLIP-RN50x4, CLIP-RN50x16), as illustrated in the tables below. These show the potential for our method to be further applied to large-scale data and better vision-language pre-trained models.
> > >
> > >
> > > Extra data | AP | AP_r |
> > > |---- | ---- |---- |
> > > None | 32.2 | 24.1
> > > CC3M-Unl | 34.0 | 24.8 (+4.6) |
> > > INet-Unl | 34.7 | 28.6 (+4.5) |
> > > INet-LVIS | 35.0 | 30.4 (+6.3)
> > >
> > > Semanatics provider | AP | AP_r |
> > > |---- | ---- |---- |
> > > None | 32.2 | 24.1
> > > CLIP-RN50 | 35.0 | 30.4 (+6.3)
> > > CLIP-RN50x4 | 36.0 | 33.0  (+8.9)
> > > CLIP-RN50x16 | 36.2 | 31.9 (+7.8)
> > >
> > > Thanks for the constructive suggestions, and we will add the discussion in our next version.
> > >
> > > [1] Zhang, Cheng, et al. "Mosaicos: a simple and effective use of object-centric images for long-tailed object detection." CVPR. 2021.
> > > [2] Li, Bo, et al. "Improving Long-tailed Object Detection with Image-Level Supervision by Multi-Task Collaborative Learning." arXiv preprint, 2022.
> > > [3] Zhou, Xingyi, et al. "Detecting twenty-thousand classes using image-level supervision." ECCV, 2022.
> > > [4] Zhong, Yiwu, et al. "Regionclip: Region-based language-image pretraining." CVPR, 2022.
> > > [5] Gu, Xiuye, et al. "Open-vocabulary Object Detection via Vision and Language Knowledge Distillation." ICLR, 2021.
> > > [6] Tan, Jingru, et al. "Equalization loss for long-tailed object recognition." CVPR, 2020.
> > > [7] Gupta, Agrim, Piotr Dollar, and Ross Girshick. "Lvis: A dataset for large vocabulary instance segmentation." CVPR, 2019.
> > > [8] Ghiasi, Golnaz, et al. "Simple copy-paste is a strong data augmentation method for instance segmentation." CVPR, 2021.
> > > [9] Kang, Bingyi, et al. "Decoupling Representation and Classifier for Long-Tailed Recognition." ICLR, 2019.

---

> > > > ### Comment · Reviewer_ndNj · 2023-08-17
> > > >
> > > > I would like to thank the authors for further clarification. I am satisfied with the discussion that the proposed method can adapt to different types of extra data (including unlabeled classification data) and various sizes of CLIP models. Thus, I am glad to promote my rating to above the ACCEPT threshold.

---

> > > > > ### Author Response · Authors · 2023-08-21
> > > > >
> > > > > Thank you for your valuable comments and suggestions. We hope that our rebuttal has effectively addressed your concerns, and it is great to see that you are “glad to promote my rating to above the ACCEPT threshold”. We just wanted to confirm that your final rating is correct in the system, since the rating remains the same before (boardline accept) and after the rebuttal. Please do not hesitate to inform us if there are any additional points you would like to see clarified or improved before the reviewer-author discussion period concludes.

---

### Official Review · Reviewer_YRik · 2023-07-06

**Soundness:** 3 good
**Presentation:** 3 good
**Contribution:** 3 good
**Rating:** 5
**Confidence:** 4

**Summary:**

This paper adopts the CLIP model to obtain a 'soft label' supervision to train the detector under the long-tail distribution dataset and derive rich semantics from the CLIP part to enhance the tail-categories representations, which can be removed during the inference. The authors claim that the CLIP model can well capture visual semantics conditioned on only coarse locations, the whole-bbox for extra data in this paper. They then elaborate soft-label supervision to train the detector so as to alleviate semantic ambiguity and location sensitivity issues. Various ablative studies have been conducted to validate the effectiveness of the proposed method.

**Strengths:**

1. The novelty of this paper sounds technically reasonable.
2. The writing of this paper is easy to follow.
3. The final performances look competitive and the improvements are obvious.

**Weaknesses:**

1. In Line 172, the subscript of f and o maybe should be superscript.
2. The proposed semantic branch training is parallel to the detection heads during the training to avoid the training conflicts for the CE loss and KL loss, but the detection heads are only trained on the tailed samples in LVIS without some kind of interactions on the semantic branch and how to ensure that the detection heads can well handle the rare categorical samples during inference?
3. ImageNet-21K is used as extra data in this paper, but the backbone is pre-trained in ImageNet too. So is there any dataset overlapping between the pertaining and fine-tuning stages?

**Questions:**

Please regard the weakness parts.

**Limitations:**

Please regard the weakness parts.

---

> ### Author Rebuttal · Authors · 2023-08-10
>
> ### q1: L172 subscripts.
> Thanks for pointing it out. We will definitely fix it in the next version.
>
> ### q2: How can semantic learning on extra classification data boost rare categories detection?
>
> The detection head is trained not only on the LVIS dataset but also on the extra image classification dataset.
>
> More specifically, we use a unified objective function $L=L_{loc}+L_{cls}+L_{soft}$ (as detailed in Section 3.3), treating both detection data and extra classification data in a unified way. Consequently, the detection head is trained using both types of datasets. For classification data without bounding box annotations, we employ pre-defined whole-image boxes as pseudo boxes and group multiple images together to create mosaics, which provide location supervision for the detection head.
>
> In conclusion, our method can help rare classes in two ways:
>
> 1) Rich semantics ($L_{soft}$): the semantic supervision flows back to the object features, implicitly enhancing the feature representation and localization capability;
> 2) Coarse locations ($L_{loc}$): our training scheme allows the detector to learn from pre-defined pseudo locations on the classification data through $L_{loc}$;
>
> To further demonstrate the effectiveness of the two perspectives, we conduct an ablation study on $L_{soft}$ and $L_{loc}$ within the training recipe for classification data. The results, shown in the table below, demonstrate that both rich semantics and coarse locations play significant roles in boosting long-tail object detection, offering significant performance gain on overall AP and rare AP.
>
> |Method | AP | AP_r | AP_c | AP_f|
> |---- | ---- |---- |---- |---- |
> | w/o $D_{extra}$ | 32.2 | 24.1 |29.9 |38.3 |
> | + $L_{soft}$ (rich semantics only) | 33.6 | 28.6 (+4.5) | 32.4 | 37.2 |
> | + $L_{loc}$ (coarse location) | 35.0 | 30.4 (+1.8) | 33.1 | 39.0 |
>
> ### q3: Data overlapping between backbone pretraining and detection training
> Yes, there is data overlap between the backbone pretraining and detection training. There are 246 categories that overlap between ImageNet-1k and LVIS, and 997 categories overlap between ImageNet-21k and LVIS.
>
> Dataset | Number of Imgs | Definition|
> |---- | ---- | ----
> |LVIS  | 0.1M | The original LVIS
> |INet-1k  | 1M | The original ImageNet-1k
> |INet-21k  | 14M | The original ImageNet-21k
> |INet-LVIS  | 1M | INet-21k classes overlapped with LVIS
>
> |Method | $D^{backbone}$ | $D^{od}$|AP | AP_r | AP_c | AP_f|
> |---- | ---- |---- |---- |---- |---- |---- |
> | w/o $D_{extra}$ | INet-1k | LVIS | 32.2 | 24.1 |29.9 |38.3 |
> | w/o $D_{extra}$ | INet-21k  | LVIS | 35.7 | 25.9 | 35.0 | 40.7
> | Ours | INet-1k  | LVIS + INet-LVIS | 35.0 | 30.4 | 33.1 | 39.0 |
> | Ours | INet-21k  | LVIS + INet-LVIS | 37.5 | 32.4 | 36.0 | 41.5 |
>
> We further conduct experiments with R50 backbones pretrained with different amounts of data under the 1$\times$ schedule. The table shows that pretraining on large-scale data can provide strong perception capability for the downstream detection task, with overall performance gain. However, the performance on rare categories is still relatively low, indicating that this approach does not alleviate the long-tail effects in detection. In addition, the pretraining cost will increase significantly with 10 times more pretraining data.
>
> In contrast, our method is more effective than pretraining to handle long-tailed detection, especially for the tail categories. Notably, our approach is still effective with strong pretrained backbones, further improving performance on long-tailed object detection.
>
> Thanks for the inspiration, and we will add the discussion on data overlapping in the next version.

---

> > ### Comment · Reviewer_YRik · 2023-08-14
> > **Rebuttal Response**
> >
> > Thanks for your detailed explanation and more experiments. In sum, most of my concerns have already been addressed. I keep my initial rating 'borderline accept'. Moreover, I suggest the authors should carefully take into account about the further demonstrations in rebuttal phase into their revised version, especially 'q2: How can semantic learning on extra classification data boost rare categories detection?'

---

### Author Rebuttal · Authors · 2023-08-10

First of all, we sincerely appreciate all your valuable comments and suggestions.

We are pleased that all reviewers think our paper is well-written and easy to follow. We are encouraged that reviewers find our proposed RichSem with reasonable novelty (YRik), significant results (ndNj), and extensive ablation and discussion (gFff, vo8U).

We carefully read the comments and attempted to provide comprehensive responses accordingly. Please find the rebuttal below each official review. We hope the responses could answer the questions raised by reviewers and address any concerns about our work.

Thanks again to all reviewers for the time and effort!

---

### Decision · Program_Chairs · 2023-09-21

**Decision:**

Accept (poster)

**Comment:**

Paper received all "accept" ratings from the reviewers post-rebuttal: 2 x Borderline Accept, 1 x Weak Accept and 1 x Accept. There is a consensus that the work is interesting, novel and the paper is well written. The approach also illustrated significant improvements on long-tailed datasets. Questions raised by the reviewers were throughly addressed in the rebuttal, including with provided (significant) additional experimental results. Reviewers were generally satisfied with the rebuttal and acknowledge that their concerns were addressed. AC has read the reviews, rebuttal, and the discussion that followed and concurs with reviewers that the work is interesting and should be accepted. Authors are encouraged to incorporate results from the rebuttal into the main paper (or supplementals) for the camera ready.